# Long-term effects of severe acute malnutrition during childhood on adult cognitive, academic and behavioural development in African fragile countries: The Lwiro cohort study in Democratic Republic of the Congo

Pacifique Mwene-Batu[1,2,3,4]*, Ghislain Bisimwa[1,3], Marius Baguma[4,5], Joelle Chabwine[4,6,7], Achille Bapolisi[1,4], Christine Chimanuka[1], Christian Molima[1], Michèle Dramaix[2], Nicolas Kashama[3], Jean Macq[8], Philippe Donnen[2]

1 Ecole Régionale de Santé Publique, Université Catholique de Bukavu (UCB), Bukavu, DR Congo, 2 Ecole de Santé Publique, Université Libre de Bruxelles, Brussels, Belgium, 3 Nutrition Department, Centre de Recherche en Sciences Naturelles de Lwiro (CRSN-Lwiro), DR Congo, 4 Hôpital Provincial General de Référence de Bukavu (HPGRB), Faculty of Medicine, Université Catholique de Bukavu (U.C.B.), Bukavu, DR Congo, 5 Neurosciences Group, Biomedical Research Institute (BIOMED), UHasselt, Diepenbeek, Belgium, 6 Laboratory of Cognitive and Neurological Science, Department of Neuroscience and Movement Science, Medicine Section, University of Fribourg, Fribourg, Switzerland, 7 Division of Neurorehabilitation, Rehabilitation Clinic, Fribourg Hospital, Fribourg, Switzerland, 8 Institute of Health and Society, Université catholique de Louvain (UCLouvain), Brussels, Belgium

* lyabpacifique@yahoo.fr

## Abstract

### Introduction

Little is known about the outcomes of subjects with a history of severe acute malnutrition (SAM). We therefore sought to explore the long-term effects of SAM during childhood on human capital in adulthood in terms of education, cognition, self-esteem and health-related disabilities in daily living.

### Methodology

We traced 524 adults (median age of 22) in the eastern Democratic Republic of the Congo, who were treated for SAM during childhood at Lwiro hospital between 1988 and 2007 (median age 41 months). We compared them with 407 community controls of comparable age and sex. Our outcomes of interest were education, cognitive function [assessed using the Mini Mental State Examination (MMSE) for literate participants, or its modified version created by Ertan et al. (MMSE-I) for uneducated participants], self-esteem (measured using the Rosenberg Self-Esteem Scale) and health-related social and functional disabilities measured using the World Health Organization Disability Assessment Schedule (WHODAS). For comparison, we used the Chi-squared test along with the Student's t-test for the proportions and means respectively.

**Data Availability Statement:** All relevant data are within the paper.

**Funding:** This study is part of a Research for Development Project entitled: 'Implementation study of a model of psycho-medico-social care at the health centre level: the case of people with chronic diseases and the mother-child malnourished couple, South Kivu, Democratic Republic of Congo' and funded by Belgian Development Cooperation through the Académie de Recherche et d'Enseignement Supérieur (ARES). JM received a grant from ARES.the funder of the study had no role in study design, data collection, data analysis, data interpretation, or writing of the report."

**Competing interests:** The authors have declared that no competing interests exist.

## Results

Compared with the community controls, malnutrition survivors had a lower probability of attaining a high level of education (p < 0.001), of reporting a high academic performance (p = 0.014) or of having high self-esteem (p = 0.003). In addition, malnutrition survivors had an overall mean score in the cognitive test that was lower compared with the community controls [25.6 compared with 27.8, p = 0.001 (MMSE) and 22.8 compared with 26.3, p < 0.001 (MMSE-I)] and a lower proportion of subjects with a normal result in this test (78.0% compared with 90.1%, p < 0.001). Lastly, in terms of health-related disabilities, unlike the community controls, malnutrition survivors had less social disability (p = 0.034), but no difference was observed as regards activities of daily living (p = 0.322).

## Conclusion

SAM during childhood exposes survivors to low human capital as regards education, cognition and behaviour in adulthood. Policy-deciders seeking to promote economic growth and to address various psychological and medico-social disorders must take into consideration the fact that appropriate investment in child health as regards SAM is an essential means to achieve this.

## Background

Malnutrition is a public health problem in low-income countries (LICs). Every year, more than 5.9 million children under the age of five die worldwide, and 45% of those deaths are attributed to undernutrition [1]. A malnourished child's risk of mortality is 11 times higher than that of a child with a good nutritional status [2]. Two million deaths a year are currently attributed to severe emaciation [3]. Mortality is even higher in cases of emaciation combined with chronic malnutrition [2]. In addition to the consequences in terms of mortality, undernutrition has serious adverse effects on physical development in terms of delayed growth and physical disability, and is associated with cognitive and behavioural disorders in children, particularly when it occurs during childhood [4–6].

In high- and medium-income countries (HMICs), it has been shown that subjects with a low birth weight and/or low weight gain in childhood are at risk of being smaller, having reduced cognitive development, a low level of education and a lower socio-economic status in adulthood, and of giving birth to underweight children, unlike children who have a good nutritional status [5, 7]. Indeed, in follow-up studies conducted in HMICs, it was observed that adults with a low birth weight and/or low weight gain during childhood scored significantly lower in various cognitive tests and attained a significantly lower level of education than adults who had no history of undernutrition. They were also at higher risk of developing dementia in old age [5, 8–10]. The reason for this is that undernutrition in childhood affects brain development and leads to less efficient brain function due to myelin deficiency, reduced dendrite branches, alteration in the function of certain neurotransmitters (serotonin and dopamine), as well as the presence of underdeveloped connectivity patterns [11, 12].

Despite growing evidence of the negative long-term effects of childhood undernutrition in HMICs on behaviour, and academic and cognitive performance [5, 8–10], little is known about the long-term outcomes of children treated for SAM in LICs. The few studies that have examined this issue are mostly old and only concern subjects who were pre-pubertal at the

time of follow-up, and in most cases, only considered growth delay during childhood as an indicator of malnutrition [6, 9, 13–15].

According to these studies, undernutrition survivors attain fewer years of education and have deficiencies in vocabulary and mathematics, along with other learning or intellectual disabilities [6, 13–15]. This poor cognitive and academic development may give them a lower probability of accessing better-paid professions, leading to reduced economic productivity in adulthood. This could create social and economic issues in disadvantaged communities that lead to the vicious circle of intergenerational poverty that hinders the economic development of the majority of LICs [5, 16–20]. However, little research has focused on the potential long-term consequences of childhood SAM on cognitive, behavioural and academic development in adulthood in LICs.

In the Democratic Republic of the Congo (DRC), malnutrition is still a major public health problem, with a chronic malnutrition prevalence of 41.8% among children under the age of five [21]. In 2017, UNICEF estimated the number of children with SAM at 1.9 million, and those with moderate acute malnutrition at 1.5 million [22]. In South Kivu, one of the 26 provinces of the DRC (located in the east of the country), malnutrition has been endemic since the 1960s [23]. Half of all children under the age of five are affected by chronic malnutrition [21].

Lwiro paediatric hospital (HPL) was one of the first facilities to be involved in treating chronic malnutrition in the DRC. A team of researchers supported by the "Centre Scientifique et Médical de l'Université Libre de Bruxelles pour ses Activités de Coopération" (CEMUBAC) developed a SAM treatment model in the 1980s, and began computerising clinical data in 1986. The electronic records contain sociodemographic, anthropometric, clinical and biological data gathered from patients hospitalised between 1988 and 2007, from the time of admission through to their discharge from hospital. Previous research studied various specific aspects of malnutrition in the DRC in general and in South Kivu in particular [24–28], but none had investigated the potentially harmful effects of SAM during childhood on long-term outcomes, particularly education, and cognitive and behavioural development in adulthood.

Our study aims therefore to evaluate the long-term effects of childhood SAM on the adult human capital of a group of young adults studied 11 to 30 years after nutritional rehabilitation, in a context of no nutrition transition (environment with a monotonous, undiversified and low-quality food situation), living in the east of the DRC. We primarily assessed the effects of childhood SAM on education, cognitive development, self-esteem, and various functional and social disabilities in adulthood.

## Methods

### Study framework

The study was conducted at the Centre de Recherche en Science Naturelle de Lwiro (CRSN-Lwiro), in the health zones of Katana and Miti-Murhesa, in South Kivu, DRC. The health zones of Miti-Murhesa and Katana are located 33 and 40 km from the city of Bukavu (capital of the province of South Kivu), respectively. The CRSN was created in 1947 and its activities are grouped under four research departments: Biology, Geophysics, Nutrition and Documentation. The Nutrition Department has a paediatric hospital and four integrated health centres which monitor the health and nutrition of children in the community. The main economic activities of the population in the health zones of Miti-Murhesa and Katana are farming, livestock, fishing and small trade [23].

### Study design and population

The initial cohort comprised 1,981 patients treated for SAM in childhood at the HPL between 1988 and 2007 [29]. On admission to the hospital, the median age was 41 months with 70.8%

of patients aged between 6 and 59 months [29]. The nutrition diagnosis made at the time (based on the weight-to-height ratio plotted on the local growth curve established by De Maeyer in 1959, the presence of nutritional oedema, and serum albumin levels) [28] was reassessed using ENA (Emergency Nutrition Assessment) Software for SMART (version October 2007) for standardisation according to the World Health Organization (WHO) child growth standards [30]. Based on the WHO standards, only 84% of the children were classed as having SAM [29]. The others, classed as moderate acute malnutrition or not suffering from acute malnutrition, were excluded. All of the patients hospitalised had been treated according to the guidelines at that time [28].

During the follow-up of these now-adult subjects, 11 to 30 years after nutritional rehabilitation, 524 subjects from the initial cohort still living in the health zones of Miti-Murhesa and Katana were surveyed [29]. These SAM survivors constitute the case group for this study. To assess their long-term growth in the follow-up study, these survivors were compared with 407 controls randomly selected from the same community [29]. The 407 subjects make up the community control group in this study. The community control was defined as a subject who had not been admitted to HPL for SAM, was the same sex, was living in the same community, and was no more than 24 months younger or older than the case. To identify these subjects, community health workers spun a bottle at the case's home, then went door to door starting with the nearest house in the direction shown by the bottle until they found a subject that met the criteria [29].

## Outcomes of interest

Our main outcomes of interest were education, cognition, self-esteem and health-related functional and social disabilities. Secondarily, we assessed access to information through the frequency with which subjects listened to the radio and used social networks.

Education was defined according to the level of education attained and their self-reported academic performance. To assess this academic performance, each participant (literate) gave his impression of his academic performance based on the last academic results obtained. This performance was divided into 3 categories: poor, average and high [31]. The self-reported academic performance was linked with the level of education attained [31, 32]. It was shown that the subjects who reported a poorer academic performance had a higher probability of abandoning education and a low level of cooperative participation in class, unlike those who considered themselves to have a higher level of academic performance [31, 32]. Additionally, this self-reported academic performance served as an indirect way for us to assess academic success independently of the level of education attained due to the fact that, with the persistent socio-economic vulnerability of households aggravated by wars in the region, the education of certain children has been compromised due to lack of financial resources.

Cognition was assessed using the Mini Mental State Examination (French consensual MMSE version of the GRECO) [33]. This test, initially developed in 1975 by Folstein and McHugh for screening dementia, is used to rapidly assess the cognitive capacities of educated subjects [34]. It comprises a series of thirty questions divided into 6 sections: orientation (10 points), registration (3 points), attention and calculation (5 points), recall (3 points), language (8 points), and repetition (1 point) [33, 34]. For uneducated subjects, Ertan et al. created a modified version (MMSE-I) [35]. For each of the thirty questions, a correct answer gave a score of 1 and an incorrect or near answer, a score of 0. The maximum score was 30. For the interpretation, a result was considered normal if the overall score was between 26 and 30 (for MMSE) and between 23 and 30 (for MMSE-I). A score below 26 for educated subjects and 23 for uneducated subjects was considered abnormal [34, 35]. The MMSE and its modified

version for illiterate subjects (MMSE-I) have been used on several occasions in HMICs with a high degree of internal consistency (Cronbach's Alpha) ranging from 0.77 for uneducated subjects in Turkey and Tunisia [36, 37] to 0.88 in uneducated Chinese subjects [8].

Self-esteem was measured using the French version of the Rosenberg Self-Esteem Scale [38]. This scale is a psychological tool developed by Morris Rosenberg, who worked particularly on self-esteem and self-concept [39]. This scale is based on ten affirmations that give a better understanding of how an individual values himself and the satisfaction he has with himself. The answers to each question were based on a scale of 1 to 4 (1: 'totally disagree', 2: 'somewhat disagree', 3: 'somewhat agree', 4: 'totally agree'). We then added the points obtained in questions 1, 2, 4, 6 and 7 (the positive questions). The score was reversed for the so-called negative questions, 3, 5, 8, 9 and 10: give 4 if you choose 1, 3 if you choose 2, 2 if you choose 3, and 1 if you choose 4. In the end, we added the totals to obtain a final score. The score obtained varied between 10 and 40. If the score was lower than 25, self-esteem was very low. Between 25 and 30, self-esteem was low. Between 31 and 34, self-esteem was average. Between 35 and 39, self-esteem was high. Lastly, if the score was 40, self-esteem was very high [38, 39]. Given that we had very few subjects in the categories 'very high self-esteem' and 'very low self-esteem' (less than 2% per group), we deemed it appropriate to regroup the scores into 3 categories for the analysis: $\leq$ 30 = low self-esteem, 31–34 = average self-esteem and $\geq$ 35 = high self-esteem. The Rosenberg Self-Esteem Scale has been validated and frequently used in low- and middle-income countries [38, 40, 41], with a high degree of worldwide internal consistency (Cronbach's Alpha) of 0.81 in 28 languages, ranging from 0.58 in the DRC (in French) to 0.85 in Tanzania (in Swahili) among students and in the community [38].

To assess health-related functional (household and professional tasks) and social (getting along with those in their immediate environment) disability, we used 2 of the components of the WHO Disability Assessment Scale (WHODAS 2.0). The WHODAS is a multidimensional and intercultural questionnaire comprising 36 items for assessing cognition, mobility, self-care, relationships with people, activities of daily living and participation in society. WHODAS 2.0 has been approved and frequently used in low- and middle-income countries [42–44], with a high degree of internal consistency (Cronbach's Alpha) ranging from 0.77 in South African women [45] to between 0.82 and 0.98 in people with several mental disorders and their caregivers in rural Ethiopia [46]. WHODAS 2.0 is also able to detect small changes over time [46]. In addition, there is a Swahili version, which was recently used in South Kivu in the east of the DRC [43].

As regards functional disabilities, we considered a subject as having: no difficulty if his score was between 6 and 10, moderate difficulty between 11 and 14, and severe difficulty between 15 and 24. Lastly, for social disabilities, depending on the score obtained, we considered a subject as having: no disability if their score was between 5 and 10, moderate disability between 11 and 15, and severe disability between 16 and 25. For the scores, the respondent judged the level of difficulty on a five-point scale (none = 1, slight = 2, moderate = 3, severe = 4, extreme/unable = 5). At the end of the questionnaire, the participant was asked to give an overall assessment of the degree of disruption to their life as regards activities of daily living and social activities according to the difficulties they had encountered in the last thirty days. For activities of daily living, there were a total of six questions. The first four questions were scored from 1 to 5, and for the last two questions, the subject obtained '1' if the answer was No and '2' for a Yes. For social disabilities, however, there were a total of five questions, all scored from 1 to 5.

To assess access to information, we used the frequency with which they listened to the radio and used social networks. To measure the frequency with which they listened to the radio, the participant chose one of the following responses: never, rarely, often or always. We then

grouped the assertions into two categories ('Yes' for answers often and always; 'No' for never and rarely). Lastly, for social network use, each participant said whether they used (Yes) or did not use (No) the most common social networks in the region (Facebook, WhatsApp, Yahoo or Gmail).

## Data collection

Data collection was carried out in two phases between January 2019 and April 2019. It was carried out by 20 trained community health workers and 2 supervising doctors and assisted by neighbourhood leaders, licensed nurses and community relays. The community health workers were the same ones that had helped identify malnutrition survivors during the creation of the cohort [29].

The first phase involved home visits. During these visits, the community health workers administered a questionnaire, which had been translated into Swahili (national language spoken in the east of the DRC), to the participants, concerning demographic data that included education and health-related functional and social disabilities. They then gave the participants an appointment 24 to 48 hours later in a health facility for the second phase.

During the second phase, a questionnaire concerning self-esteem (Rosenberg Self-Esteem Scale) and cognition (MMSE and MMSE-I) were administered to the participants.

The general questionnaire contained variables relating to the participant's identity, education, self-reported academic performance, cognitive function assessed using the MMSE and MMSE-I tests, self-esteem (Rosenberg Self-Esteem Scale), and health-related daily functional and social disabilities. The final data concerned the frequency with which they listened to the radio and used social networks.

The basic structured questionnaire in French was translated into Swahili according to a rigorous translation protocol to ensure intercultural and conceptual equivalence. A French-speaking translator from the language school of the Université Catholique de Bukavu, and whose first language is Swahili, carried out the translation. A bilingual panel composed of the principal researcher, key health professionals working in the field of health in the two zones covered by our study, and community health officers examined the translated version in order to correct any intercultural issues in terms of incomprehensibility or lack of clarity. The resulting document was then translated into French by an independent bilingual reviser. Once translated, a native French speaker compared this French back-translation with the original French version to confirm its equivalence. Before proceeding with the study, itself, the questionnaire was revised and modified following tests in the field, and a pilot study was conducted with a subgroup of 30 people from the region, selected from within the community. The interviews lasted between 30 and 45 minutes.

As regards the cognition assessment using the MMSE and its modified version (MMSE-I), scores were entered directly in the grid at the time of the assessment. Literacy was assessed on the basis of the level of education attained. Subjects who had not completed primary education were considered illiterate. Due to copyright on the use of the MMSE [47, 48] and our limited means, we only used it in a subgroup. To select them, we took subjects who were older than 25 years old and of comparable sex, regardless of their level of education. According to this criterion, out of the initial 524 cases, only 100 were selected. A total of 100 individuals aged more than 25 years, out of the initial 407, were selected as controls. This threshold was used because, beyond the age of 25, humans start to lose their first neurons and a slight decrease in cognitive performance can be observed: decline in memory efficiency, slight slowing of thought, more difficult perception and slower perceptual identification, etc. [49]. In addition, at age 25, an individual is expected to have completed his or her education in our context.

## Statistical analysis

We used Stata version 13.1 software for our analyses. The size of the sample was determined by the number of patients admitted for SAM to the Lwiro paediatric hospital from 1998 to 2007 and living in Miti-Murhesa and Katana in 2018. Categorical variables were summarised in the form of frequency and proportion. Quantitative data were presented in the form of a mean and standard deviation (SD) or a median and minimum-maximum (min-max) depending on whether the distribution was symmetrical. Pearson's Chi-Squared Test (or Fischer's Exact Test if the criteria for the use of the Chi-Squared were not met) was used to compare the category variables, and the Student t-test to compare the mean between the two groups.

## Ethical considerations

This study was approved by the Institutional Review Board of the Université Catholique de Bukavu, and informed consent was obtained from all participants. Respondents provided signed informed consent for participation in the study, either by written signature or by fingerprints, depending on literacy. Child assent was obtained for respondents below 18 years of age, after a parent or guardian's consent. All procedures performed were in accordance with the ethical standards of the institutional ethical committee and with the 1964 Helsinki declaration and its later amendments.

# Results

## Sociodemographic characteristics

Table 1 shows the sociodemographic characteristics of our two groups. The median age in the two groups was 22, and males accounted for 52.1% and 50.6% of the cases and controls respectively. Nearly all of our subjects were from the Shi ethnic group in both groups. Approximately

**Table 1. Sociodemographic and economic characteristics of the population under study.**

|  | Cases | | Community Controls | | |
|---|---|---|---|---|---|
|  | N (total) | % | N (total) | % | p-value |
| **Age (year)** |  | 22 (16–40) * |  | 22 (16–40) * | 0.541 |
| **Male** | 524 | 52.1 | 407 | 50.6 | 0.341 |
| **Civil status** |  |  |  |  |  |
| Living alone |  | 54.2 |  | 62.1 |  |
| Living in a couple |  | 45.8 |  | 37.9 | 0.129 |
| **Tribe** | 515 |  | 403 |  |  |
| Shi |  | 99.0 |  | 98.2 |  |
| Other |  | 1.0 |  | 1.8 | 0.879 |
| **Religion** |  |  |  |  |  |
| Catholic |  | 44.8 |  | 45.2 |  |
| Protestant |  | 50.3 |  | 52.1 | 0.798 |
| Other |  | 4.9 |  | 2.7 |  |
| **Occupational category** | 479 |  | 359 |  |  |
| Management |  | 3.1 |  | 7.5 |  |
| Administrative + office worker |  | 0.8 |  | 1.1 | 0.137 |
| Farmer + fisher + market vendor |  | 64.9 |  | 62.1 |  |
| Unskilled workers |  | 31.1 |  | 29.2 |  |

*Data are Median (Min–Max).

two thirds of our population sample were from occupational category three (farmer, fisher, market vendor).

## Education, cognitive function and self-esteem

Differences in terms of education, cognitive function and self-esteem between the cases and the community controls are shown in Table 2.

As regards education, the level attained was significantly lower in the cases compared with the controls. Self-reported academic performance was also significantly lower in the cases compared with the controls.

Comparing the cognitive function of the two groups using the MMSE and MMSE-I, we noted that the global mean scores were significantly lower in the cases compared with the controls. The proportion of individuals who had a normal test was significantly lower in the cases compared with the controls.

Overall, the cases had statistically significant lower self-esteem than the controls.

Lastly, compared with the controls, a significantly lower proportion of cases regularly listened to the news on the radio or used social networks.

## Health-related disabilities in the past 30 days

In terms of health-related disabilities in the past 30 days, summarised in Table 3, we observed no difference in disability in carrying out activities of daily living between the two groups. However, the cases had significantly less disability in terms of social relationships than the community controls.

**Table 2. Difference in education, self-esteem and cognition between cases and controls.**

| | Cases | | | Community Controls | | | p-value |
|---|---|---|---|---|---|---|---|
| | N (total) | % | Mean (SD) | N (total) | % | Mean (SD) | |
| *EDUCATION | | | | | | | |
| 1. Level of education | 524 | | | 407 | | | |
| None | | 27.8 | | | 20.0 | | |
| Primary | | 37.1 | | | 33.6 | | < 0.001 |
| Secondary | | 34.2 | | | 42.0 | | |
| University | | 1.0 | | | 4.4 | | |
| 2. SR academic performance | 378 | | | 325 | | | |
| Low | | 23.8 | | | 15.2 | | |
| Average | | 45.1 | | | 49.0 | | 0.014 |
| High | | 31.0 | | | 35.8 | | |
| *COGNITION | 100 | | | 100 | | | |
| Mean MMSE score (SD) | | 50.0 | 25.6 (2.6) | | 72.0 | 27.8 (2.2) | 0.001 |
| Mean MMSE-I score (SD) | | 50.0 | 22.8 (2.1) | | 28.0 | 26.3 (2.9) | < 0.001 |
| Normal | | 78.0 | | | 90.1 | | < 0.001 |
| *SELF-ESTEEM | 518 | | | 405 | | | |
| Low | | 20.5 | | | 12.1 | | |
| Average | | 72.6 | | | 78.5 | | 0.003 |
| High | | 6.9 | | | 9.4 | | |
| *Listens to the news | | | | | | | |
| Listens to the radio news (yes) | | 40.0 | | | 49.2 | | 0.007 |
| Uses social networks | | 14.0 | | | 21.7 | | 0.003 |

**Table 3. Difference in terms of health-related disabilities between cases and controls.**

| | Cases | | Community Controls | | p-value |
|---|---|---|---|---|---|
| | N (total) | % | N (total) | % | |
| **HEALTH-RELATED DISABILITIES** | | | | | |
| **1. Activities of daily living** | **523** | | **401** | | |
| No disability | | 71.3 | | 68.6 | |
| Moderate disability | | 16.4 | | 17.0 | 0.322 |
| Severe disability | | 12.2 | | 14.5 | |
| **2. Social aspects** | 521 | | 398 | | |
| No disability | | 94.2 | | 91.0 | |
| Moderate disability | | 5.0 | | 8.8 | 0.034 |
| Severe disability | | 0.8 | | 0.3 | |

## Discussion

The purpose of our study was to explore the long-term effects of childhood SAM on, primarily, education, cognitive function, self-esteem and health-related social and functional disabilities in adulthood in the context of the DRC, and secondarily, on information seeking.

Our results suggest that SAM during childhood has long-term negative effects on cognitive function and exposes survivors to being less competitive academically as well as having lower self-esteem. Nevertheless, in terms of health-related disabilities, they have less disability on a social level than the general population, but no difference was observed as regards activities of daily living. Lastly, SAM survivors listen to the news less and use social networks infrequently compared with the controls.

To our knowledge, our study is the first in an LIC to assess the long-term effects of childhood SAM in adults after a long period (between 11 and 30 years), after discharge from hospital in a context of endemic malnutrition and crisis. Our study is original in that it: 1) assesses self-esteem, self-reported academic performance, information seeking, and health-related disabilities in addition to education and cognitive functions, 2) has a large sample size, and 3) considers SAM as an exposure.

As regards education and cognitive function, we observed that compared with the community controls, malnutrition survivors were less likely to attain a high level of education, and had a lower mean score on the cognitive test and a higher proportion of subjects with an abnormal cognitive test result. Our findings are consistent with those of several other studies that have shown a strong link between undernutrition during childhood and reduced human capital in later life as regards education and cognitive function [5, 6, 16, 17]. One reason for this is that undernutrition during childhood has an irreversible negative impact on brain development due to a decrease in myelin, an increase in neuronal mitochondria, a decrease in cortical dendrites in the neural spines and a lower ratio of granule cells to Purkinje cells in the cerebellum, altering motor and cognitive development [50]. In addition to SAM, 94% of cases suffered delayed growth in childhood [29]. It has been shown that delayed growth is associated with altered hypothalamic-pituitary-adrenal activity with high levels of cortisol, a fast heart rate and a high level of urinary epinephrine, which can also lead to reduced cognitive abilities in school-age children [51]. In addition, children with malnutrition do not have sufficient energy or the essential motor skills to thrive at school and therefore complete fewer years of education, which in turn affects their cognition in adulthood [52]. Lastly, malnutrition survivors mostly return to the same disadvantaged conditions after their nutritional rehabilitation [53]. As such, in addition to the effects of malnutrition on academic performance, there are

precarious economic conditions which mean that parents cannot pay the school fees to send their children to school [54].

Another relevant point is the inverse relationship between education and cognitive disorders in adulthood measured using cognitive tests [55]. Indeed, it has been shown that education in early life promotes brain growth over the years of learning and helps the brain network to function more efficiently, offering protection against cognitive decline later in life [56]. Moreover, higher levels of education often lead to professions that involve cognitive challenges and practices, which could further improve or maintain the cognitive reserve in adulthood [57]. Consequently, given that malnutrition survivors are less likely to attain a high level of education, but also due to the fact that brain alterations become more significant as age advances [49], we believe that these malnutrition survivors could run a greater risk of developing senile dementia than the general population, as has been observed in other studies [8, 50, 57]. As such, tackling childhood malnutrition could have an impact on preventing dementia in LICs. It is therefore important to initiate early neurodevelopmental follow-up of SAM survivors in order to screen for any difficulties early on and to refer such cases to the appropriate health facilities.

In terms of self-esteem and self-reported academic performance, our results show that malnutrition survivors have a higher probability of having lower self-esteem as adults and of reporting poor academic performance in relation to their community controls. We believe that one reason for this is the fact that SAM is stigmatising in society. In addition, the occurrence of SAM is often linked with a precarious socio-economic status in Africa and, unfortunately, the majority of these subjects return to the same disadvantaged conditions after nutritional rehabilitation [8, 50, 57], which has a negative impact on education. Later, this lower level of education reduces the likelihood of accessing better-paid professions, exposing these subjects to precarious living conditions, to lower household spending and, lastly, a greater risk of poverty [16]. All this leads malnutrition survivors to have lower self-esteem than the general population, in addition to the poor academic performance they self-report.

Lastly, by comparing malnutrition survivors with community controls in terms of health-related disabilities in daily living (both on the social and functional level), we observed that malnutrition survivors had relatively few social problems in terms of getting on with those in their immediate environment, than their controls, contrary to our hypotheses. This suggests that malnutrition survivors are probably more resilient than their controls as they need social bonds due to the stigmatising nature of SAM in the community. Furthermore, community controls have relatively high self-esteem, boosted by a higher level of education and consequently have a tendency to believe they have the monopoly over knowledge. This would probably create much more problems for them in the community, unlike malnutrition survivors who may tend to be more submissive, and to be less involved in discussions due to the fact that they listen to the news less and use social networks less. Consequently, they are more attentive to what others say, do not draw attention to themselves due to their low self-esteem, and are therefore less likely to have social problems. However, on the functional level, no difference was observed. This is consistent with the findings of studies conducted in HMICs, which observed that undernutrition in childhood was not associated with disabilities in carrying out daily tasks (household and professional) in adulthood [58, 59]. This may be due to the benefit of nutritional rehabilitation and catch-up growth after treatment, promoting the social reintegration of malnutrition survivors by making them autonomous in daily life, despite their cognitive disabilities.

However, certain limitations in relation to our findings are worth mentioning. Firstly, our biggest limitation is the bias of 'lost-to-follow-up'. Indeed, only 524 of the 1,981 subjects from the initial cohort were studied, and they may have different characteristics to those of the

subjects not studied. It is therefore difficult for us to form an opinion on the impact of their data on negative long-term effects given that we do not have any information on their outcomes. Nevertheless, we estimate that this would not be significantly different to our main findings given that the characteristics on admission do not differ between those lost to follow-up and the subjects who were traced [29].

Secondly, the lack of information on the social and environmental conditions in which the participants grew up is another important limitation of the study. The study design is incapable of separating mechanisms due to SAM per se from mechanisms due to persistent effects of the child's early environment or persistent living in the same poor environment.

Thirdly, there is uncertainty as to whether all of the control recruited were in good health. Although they did not present with Kwashiorkor and were not treated for SAM, some of them may have presented with moderate malnutrition linked to unfavorable socio-economic conditions in the region, but not to the point of being admitted to a hospital. The constant unfavorable situation in which the two groups lived may have substantively affected any inter-group differences.

Fourthly, we used psychometric tests (MMSE and MMSE-I) that have not been validated in the Congolese population. Consequently, we do not know how much of an impact this had on the results. Nevertheless, having used a rigorous process with pre-tests and a pilot study to correct possible intercultural issues in terms of incomprehensibility or lack of clarity, we do not think that the lack of validation will have significantly altered our main conclusions.

Fifthly, we do not have demographic information concerning infancy, including gestational age, birth weight and height, rate of growth in the first two years of life, episodes of infectious diseases (particularly diarrhoeal) as well as socio-economic data concerning their mothers (level of education and age on birth). These items of information could be potential confounding factors, because they are linked with both malnutrition and negative long-term effects on education and cognition [60].

Sixthly, we have not assessed iron, folate and iodine status in our study. Given that SAM is associated with micronutrient deficiencies which are also risk factors for cognitive disabilities [60], we cannot exclude the possible effects of these deficiencies on the different cognitive disorders.

To conclude, survivors of childhood SAM have reduced human capital later in life in terms of education, self-esteem and cognitive function. However, they have less social disabilities in daily life and their daily activities are not compromised. Policy-deciders and funders seeking to promote economic growth and combat various complex psychological and medico-social disorders must be aware of the fact that appropriate investment in child health as regards SAM is an essential means to reducing the burden associated with extreme poverty and with the health costs of psychological and medico-social disorders.

## Acknowledgments

We are thankful to Professor Philippe Hennart and the CEMUBAC team who initiated the system for collecting and computerizing data on malnutrition at Lwiro Hospital since 1988. Our thanks also go to the entire team of doctors and nurses who participated in the treatment of malnourished children between 1988 and 2007. We also thank the Van Buren Foundation which used to contribute to the functioning of Lwiro Hospital. Finally, we express gratitude to all community health workers for the data collection and administrative support during this work. We appreciate the support of Sud-Kivu Health authorities, CRSN authorities, and all village leaders involved in this study.

## Author Contributions

**Conceptualization:** Pacifique Mwene-Batu, Ghislain Bisimwa, Jean Macq, Philippe Donnen.

**Data curation:** Pacifique Mwene-Batu, Christine Chimanuka, Nicolas Kashama.

**Formal analysis:** Pacifique Mwene-Batu.

**Funding acquisition:** Ghislain Bisimwa, Jean Macq.

**Investigation:** Pacifique Mwene-Batu, Philippe Donnen.

**Methodology:** Pacifique Mwene-Batu, Michèle Dramaix, Philippe Donnen.

**Project administration:** Jean Macq.

**Software:** Pacifique Mwene-Batu.

**Supervision:** Pacifique Mwene-Batu, Michèle Dramaix, Jean Macq, Philippe Donnen.

**Validation:** Ghislain Bisimwa, Michèle Dramaix, Philippe Donnen.

**Visualization:** Pacifique Mwene-Batu, Michèle Dramaix.

**Writing – original draft:** Pacifique Mwene-Batu.

**Writing – review & editing:** Ghislain Bisimwa, Marius Baguma, Joelle Chabwine, Achille Bapolisi, Christine Chimanuka, Christian Molima, Michèle Dramaix, Philippe Donnen.

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
