## [Decision Letter · Decision Letter 0]

23 Nov 2020

PONE-D-20-21141

Long-term effects of Severe Acute Malnutrition during childhood on adult cognitive, academic and behavioural development in African fragile countries: The Lwiro Cohort Study in Democratic Republic of the Congo

PLOS ONE

Dear Dr. Mwene-Batu,

Thank you for submitting your manuscript to PLOS ONE. After careful consideration, we feel that it has merit but does not fully meet PLOS ONE’s publication criteria as it currently stands. Therefore, we invite you to submit a revised version of the manuscript that addresses the points raised during the review process.

We look forward to receiving your revised manuscript.

Kind regards,

Frédéric Denis, Ph.D.

Academic Editor

PLOS ONE

Journal Requirements:

Reviewers' comments:

Reviewer's Responses to Questions

**Comments to the Author**

1. Is the manuscript technically sound, and do the data support the conclusions?

Reviewer #1: Yes

2. Has the statistical analysis been performed appropriately and rigorously? 

Reviewer #1: Yes

3. Have the authors made all data underlying the findings in their manuscript fully available?

Reviewer #1: Yes

4. Is the manuscript presented in an intelligible fashion and written in standard English?

Reviewer #1: Yes

5. Review Comments to the Author

Reviewer #1: The present manuscript summarises the findings of a retrospective case control study in the DRC. Despite the study limitations, the paper is a useful addition to the current literature and provide unique insights for public health from an understudied population.

The lack of information on the social and environmental conditions in which the participants grew up is an important limitation of the study. It is quite likely that the cases grew up in more disadvantaged child caring, social and living conditions which predisposed them to adverse development outcomes later in life. Although this is acknowledged briefly as a study limitation, I would like to see a stronger discussion on these aspects. For instance, did you assert that the controls grew up in the same villages as the cases? Did you assess malnutrition history in the controls? Please discuss possible bias.

In addition, more detail is needed regarding the selection of the control. “407 controls randomly selected from the community”. Confirm that this is the same community. In addition, explain how the random selection was done. There seems to be discrepancy with the sampling approach described in the next lines (which indicate not a random, but a systematic sampling).

Selection of subsample of the 200 participants for the assessment of the MMSE. Did you do this randomly? Are these both sub-samples also comparable in terms of socio-demographic conditions.

The findings of the analysis are essentially based on statistical difference. A discussion regarding the practical significance of these difference (vs the bias of assessment) would be informative.

Minor

Verify use of acronyms (and avoid where possible) Eg. Abstract: explain DRC, Chronic malnutrition (CM) abbreviated twice

Reference style needs to be consistent and verified (some first names are mentioned, some last names are all caps etc

Line 390 and following “This could be explained by the fact that” or “Another explanation could be the fact that” , “This could probably be explained by” could be removed to improve readability

6. PLOS authors have the option to publish the peer review history of their article (what does this mean?). If published, this will include your full peer review and any attached files.

Reviewer #1: **Yes: **Carl Lachat

---

## [Author Response · Author response to Decision Letter 0]

10 Dec 2020

Dear Sir,

 We submit the revised version of our manuscript entitled “Long-term effects of Severe Acute Malnutrition during childhood on adult cognitive, academic and behavioural development in African fragile countries: The Lwiro Cohort Study in Democratic Republic of the Congo” [PONE-D-20-21141] by Pacifique Mwene-Batu et al.

First of all, we would like to thank the Reviewer for his very insightful comments that helped improving the quality of the manuscript. We implemented his suggestions and comments, and provide below detailed answers: 

Answers to reviewer

Reviewer #1: The present manuscript summarises the findings of a retrospective case control study in the DRC. Despite the study limitations, the paper is a useful addition to the current literature and provide unique insights for public health from an understudied population.

1) The lack of information on the social and environmental conditions in which the participants grew up is an important limitation of the study. It is quite likely that the cases grew up in more disadvantaged child caring, social and living conditions which predisposed them to adverse development outcomes later in life. Although this is acknowledged briefly as a study limitation, I would like to see a stronger discussion on these aspects. For instance, did you assert that the controls grew up in the same villages as the cases? Did you assess malnutrition history in the controls? Please discuss possible bias.

Answer: We thank the Reviewer for this pertinent remark and we accordingly have added some aspects in terms of limitations in the discussion to raise some points that were not clearly elucidated long before. 

Please see Page 12, lines 436-444: “Secondly, the lack of information on the social and environmental conditions in which the participants grew up is another important limitation of the study. the study design is incapable of separating mechanisms due to SAM per se from mechanisms due to persistent effects of the child's early environment or persistent living in the same poor environment.

Thirdly, there is uncertainty as to whether all of the control recruited were in good health. Although they did not present with Kwashiorkor and were not treated for SAM, some of them may have presented with moderate malnutrition linked to unfavorable socio-economic conditions in the region, but not to the point of being admitted to a hospital. The constant unfavorable situation in which the two groups lived may substantively affected any inter-group differences”

2) In addition, more detail is needed regarding the selection of the control. “407 controls randomly selected from the community”. Confirm that this is the same community. In addition, explain how the random selection was done. There seems to be discrepancy with the sampling approach described in the next lines (which indicate not a random, but a systematic sampling).

R/ Thanks for the suggestion. 

We confirm that controls were from the same community as cases. We have clarified this in the revised version of manuscript. See page 3, lines 149-150: “…these survivors were compared with 407 controls randomly selected from the same community [29]”.

The random selection of the control group has been explained in the manuscript, on page 3 lines 151-155: “The community control was defined as a subject who had not been admitted to HPL for SAM, was the same sex, was living in the same community, and was no more than 24 months younger or older than the case. To identify these subjects, community health workers (CHWs) spun a bottle at the case’s home, then went door to door starting with the nearest house in the direction shown by the bottle until they found a subject that met the criteria [29]”.

3) Selection of subsample of the 200 participants for the assessment of the MMSE. Did you do this randomly? 

R/ Thanks for the question.

As mentioned in the manuscript, we limited the cognitive function evaluation (using the MMSE) only 200 participants because of financial reasons. In fact, the MMSE test is copyrighted. Clinicians and researchers have to pay for its usage. Due to limited funds, we could afford only the cost of 200 tests.

To choose the 200 participants (including 100 cases and 100 controls), we first decided to include only participants aged 25 years and above. This threshold was chosen as most subjects are supposed to have completed their studies at this age. Furthermore, it has been reported that some changes occur in neurons in terms of aging, and these changes start beyond the age of 25.

Out of a total of 931 participants (524 cases and 407 controls) aged 25 years and above, 200 were selected as follows:

1°. Firstly, the reduced choice of subjects is explained by the fact that the use of the MMSE is paid for when our means were limited, which is why we wanted to do it in a subgroup.

2°. Second, in terms of subgroup identification, we thought it would be best to consider an age at which subjects would have already completed most of the studies and at which certain changes in neurons occur in terms of aging. That is why we took into account the age of 25 years.

3°. Finally, taking into account age 25 years and above as inclusion criteria in this subgroup for the MMSE, we identified 100 cases among the 524 at baseline (identified) as well as 100 controls among the 407 (randomly identified at baseline), of comparable age and sex and living in the same community.

This was explained in the revised manuscript of line 273 to 281.

“Due to copyright on the use of the MMSE [47,48] and our limited means, we only used it in a subgroup. To select them, we took subjects who were older than 25 years old and of comparable sex, regardless of their level of education. According to this criteria, out of the initial 524 cases, only 100 were selected. A total of 100 individuals aged more than 25 years, out of the initial 407, were selected as controls. This threshold was used because, beyond the age of 25, humans start to lose their first neurons and a slight decrease in cognitive performance can be observed: decline in memory efficiency, slight slowing of thought, more difficult perception and slower perceptual identification, etc. [49]. In addition, at age 25, an individual is expected to have completed his or her education in our context”.

4) Are these both sub-samples also comparable in terms of socio-demographic conditions.

R/Thanks for the question.

With regard to the socio-demographic characteristics of the different individuals in the subgroup, prior analyses were done to see if there were any differences in education between the two subgroups that could justify the observed difference in scores. After analysis, no difference in educational attainment was observed between cases and controls for the educated. For the uneducated, this was not necessary.

Below are the results comparing education between cases and controls for the educated.

 Cases Community Controls p-value

 N (total) % N (total) % 

Level of education 50 72 

 None 

 Primary 38.0 36.1 �

 Secondary 62.0 59.7 

 University 0.0 4.2 

Minor

Verify use of acronyms (and avoid where possible) Eg. Abstract: explain DRC, Chronic malnutrition (CM) abbreviated twice.

Reference style needs to be consistent and verified (some first names are mentioned, some last names are all caps etc.

Line 390 and following “This could be explained by the fact that” or “Another explanation could be the fact that”, “This could probably be explained by” could be removed to improve readability.

R/ Thanks for the remarks. 

All comments, suggestions and instructions have been implemented into the revised version.

---

## [Editor Report · Decision Letter 1]

11 Dec 2020

Long-term effects of Severe Acute Malnutrition during childhood on adult cognitive, academic and behavioural development in African fragile countries: The Lwiro Cohort Study in Democratic Republic of the Congo

PONE-D-20-21141R1

Dear Dr. Mwene-Batu,

We’re pleased to inform you that your manuscript has been judged scientifically suitable for publication and will be formally accepted for publication once it meets all outstanding technical requirements.

Kind regards,

Frédéric Denis, Ph.D.

Academic Editor

PLOS ONE
---

## [Editor Report · Acceptance letter]

16 Dec 2020

PONE-D-20-21141R1 

Long-term effects of Severe Acute Malnutrition during childhood on adult cognitive, academic and behavioural development in African fragile countries: The Lwiro Cohort Study in Democratic Republic of the Congo 

Dear Dr. Mwene-Batu:

I'm pleased to inform you that your manuscript has been deemed suitable for publication in PLOS ONE. Congratulations! Your manuscript is now with our production department. 

Kind regards, 

on behalf of

Dr. Frédéric Denis 

Academic Editor

PLOS ONE